# Root treatment with oxathiapiprolin, benthiavalicarb or their mixture provides prolonged systemic protection against oomycete foliar pathogens

**Yigal Cohen** [ID] *

The Mina & Everard Goodman Faculty of Life Sciences, Bar-Ilan University, Ramat-Gan, Israel

* ycohen@biu.ac.il

**Data Availability Statement:** All relevant data are within the manuscript.

**Funding:** The authors received no specific funding for this work.

## Abstract

Oxathiapiprolin is a fungicide effective against downy mildews of cucumber (*Pseudoperonospora cubensis*) and basil (*Peronospora belbahrii*) and late blight of tomato (*Phytophthora infestans*). To avoid fungicide resistance, it is recommended to apply oxathiapiprolin as a mixture with a partner fungicide that have a different mode of action. Here it is shown that a single application of oxathiapiprolin, benthiavalicarb, or their mixture (3+7, w/w) to the root of nursery plants grown in multi-cell trays provided prolonged systemic protection against late blight and downy mildews in growth chambers and in field tests. Soil application of 1mg active ingredient per plant provided durable protection of up to four weeks in tomato against late blight, cucumber against downy mildew and basil against downy mildew. Not only did the mixture of oxathiapiprolin and benthiavalicarb provide excellent systemic control of these diseases but also mutual protection against resistance towards both oxathiapiprolin and benthiavalicarb.

## Introduction

Oxathiapiprolin is a piperidinyl-thiazole-isoxazoline fungicide (FRAC code 49) [1, 2] with extremely high activity against plant pathogenic oomycetes [3–11] except Pythium species [2]. Oxathiapiprolin is effective when applied preventatively or curatively to the foliage or to the root system [12,13] and shows translaminar and acropetally systemic translocation [2, 3, 5,11].

The molecular target of oxathiapiprolin is the oxysterol binding protein (OSBP) [2,14], a member of the OSBP-related proteins family of lipid transfer proteins [15]. The target proteins of oxathiapiprolin in *Phytophthora capsici* and *Pseudoperonospora cubensis* genome were annotated, but their function was not disclosed[5, 16].

Oxathiapiprolin is a high-risk fungicide [5, 12, 14, 16], which requires careful use in the field to avoid buildup of resistant mutant isolates [17, 18].

Available oxathiapiprolin-based fungicides include mixtures with chlorothalonil (Orondis-Opti), mandipropamid (Orondis-Ultra), azoxystrobin, mefenoxam (Orondis-Gold), famoxadone (Zorvec Encantia) or benthiavalicarb (Zorvec-Endavia). We have recently reported on

**Competing interests:** The authors have declared that no competing interests exist.

synergistic efficacy of all mixtures, except that with benthiavalicarb, against late blight in tomato [9] and downy mildew in cucumber [4].

The present study reports on the efficacy of oxathiapiprolin+benthiavalicarb (Zorvec-Endavia [ZE]) in controlling foliar oomycete pathogens. Because both compounds are systemic, the systemic protective efficacy of ZE when applied to the root system by soil drench was also examined.

Bentiavalicarb (together with mandipropamide, dimethomorph, flumorph and iprovalicarb) belongs to the carboxylic acid amides (CAA) group of fungicides (FRAC code 40). Their mode of action involves inhibition of cell wall synthesis of oomycetes by blocking the activity of cellulose synthase *Ces*3A of the pathogen. They are effective against *Phytophthora infestans* [19], *Plasmopara viticola* [20, 21], *Pseudoperonospora cubensis* [22] and *Bremia lactucae* [23]. Resistance against CAAs occurs in field populations of *P. viticola* [21] and *P. cubensis* [24] but not in *P. infestans* or *B. lactucae* [25]. With *P. infestans*, we were able to artificially mutate sporangia for stable resistance to the phenylamide fungicide mefenoxam but failed to select mutants with stable resistance against CAAs [26], suggesting a low risk of resistance developing in this pathogen in field populations against CAAs.

Soil treatment with oxathiapiprolin was reported to be effective against soil borne oomycete pathogens, e.g., black shank in tobacco caused by *Phytophthora nicotianae* [12] and root rots in citrus caused by Phytophthora species [13]. In our recent papers, we reported that root treatment with oxathiapiprolin or oxathiapiprolin+mefenoxam were effective in controlling downy mildew in cucumber [4] and late blight in tomato [9]. The half-life time of oxathiapiprolin in soil is about a week [27, 28].

The objective of the present research was to examine the efficacy of root treatment with oxathiapiprolin, benthiavalicarb or their mixture oxathiapiprolin+benthiavalicarb (Zorvec Endavia) in controlling downy mildew in cucumber and basil and late blight in tomato in growth chambers and the field.

## Materials and methods

### Fungicides

Oxathiapiprolin (OXPT) 100OD (oil dispersion) was a gift from DuPont, France. Benthiavalicarb (BENT) 98% technical grade was a gift from Syngenta Crop Protection, Switzerland. The mixture of OXPT+BENT (Zorvec Endavia 100SC = ZE, 30g oxathiapiprolin+70g benthiavalicarb/Kg) was a gift from Agrochem Ltd (Petah Tikva, Israel). The fungicides (except BENT) were suspended in water and diluted to a series of 10-fold concentration suspensions from 0.001 to 1 mg ai per ml. BENT was dissolved in DMSO (dimethylsulfoxide, 100 mg/10 ml) and then suspended in water. For the dual mixtures, an indicated dose represents the combined doses of both ingredients.

### Plants

Cucumber (*cv* Nadyojni, own production) and tomato (*cv* Roter Gnom, deterministic growth type, a gift from Syngenta, Switzerland, or *cv* 129, non- deterministic type, a gift from Hazera Seeds, Israel) were used. Plants were grown from seeds in the greenhouse, in either 250 ml pots filled with peat: perlite (10:1, v/v) soil, one plant per pot or in multi-cell trays (cell size 5x5 cm, Hishtil, Petah Tikva, Israel) filled with the same soil. Cucumber plants were used at the 1-leaf stage, unless stated otherwise. Tomato plants were used at the 3-leaf stage, unless stated otherwise. Basil plants (*cv* Perri, a gift from Volcani Center, Newe Yaar, Israel) were grown in multi-cell trays (cell size 2.5x2.5 cm, Hishtil, Petah Tikva, Israel). Plants that grew in multi-cell trays were transplanted to 250 ml pots or to the field as indicated below.

## Root treatment

A single aliquot (1 ml) of OXPT, BENT or ZE was applied to the root (soil drench) of cucumber, tomato or basil plants. Tomato plants growing in 250 ml pots were treated at the 5, 8, 10 or 12-leaf stage. Cucumber plants growing in similar pots were treated at 1, 2, 3 or 6-leaf stage. Basil plants growing 250 ml pots were treated at the eight or 10-leaf stage. Plants growing in multi-cell trays were treated at the 1-leaf stage (cucumber), 2-leaf stage (basil) or 3-leaf stage (tomato) by pipetting 1 ml fungicide suspension of 10, 100 or 1000-ppm a.i. (0.01, 0.1 or 1 mg ai/plant) onto the soil surface around the stem base of each plant (Fig 1). In some experiments, 0.3, 0.7 or 1 ml of 10, 100 or 1000-ppm a.i. of OXPT, BENT or ZE, respectively, were applied to the root of each plant. At 1h and 5h after soil drench, 2 ml of water was added to the soil surface of each treated plant to facilitate the uptake of the fungicide(s) by the root system.

## Pathogens and inoculation

Isolate 260 of *Pseudoperonospora cubensis* (collected from cucumbers in Ahituv, Israel in May 2018) was used for inoculation of cucumber in growth chamber studies. This isolate is resistant to mefenoxam (MFX) and belongs to the A1 mating type. Isolate 164 of *Phytophthora infestans* (collected in March 2016 from potato at Nirim, Western Negev, Israel) was used for inoculation of tomato plants in growth chambers. This isolate is resistant to MFX and belongs to genotype 23_A1. Isolate Knafo 3 of *Peronospora belbahrii* was used to inoculate basil plants in growth chamber studies. This isolate was collected in 2013 at Ein Tamar, Southern Jordan valley, Israel; it is resistant to MFX.

## Growth chamber experiments

Root-treated and control untreated plants growing in 250 ml pots were spray-inoculated with spore suspension ($5x10^3$ sporangia/ml) of their respective pathogen at various time intervals after soil drench (see below). Disease records were taken at 7dpi (days post inoculation), unless stated otherwise. In cucumber, the number of downy mildew lesions produced in each plant (or leaf area infected) was recorded [4]. In tomato, the proportion of blighted leaf area was visually estimated [9]. In basil, the proportion of infected leaf area was estimated as described before [29].

## Field experiments

Experiments were conducted during 2019 at Bar-Ilan University Farm (32Ê 04' 2.40" N, 34Ê 50' 19.79" E) in 50x6 m net houses covered with white, 50 mesh, insect-proof plastic screens. Root-treated and untreated control cucumber and basil plants were transplanted to the field, at one or more days after treatment (DAT), where they were subjected to natural infection with *P. cubensis* or *P. belbahrii*. Disease records were taken upon appearance of the first symptoms of downy mildew and continued at intervals of 1–4 days for 3–4 weeks after transplantation.

## Data analysis

Eight growth chamber experiments were conducted with 4–20 replicate plants per treatment. Effective doses (ED90 values, the dose of fungicide required to inhibit disease by 90% relative to control) were derived from log-probit regression curves using SPSS software. SF (synergy factor) was calculated using the Abbott or Wadley formulae as described before [30, 31]. T-test was performed using SPSS software to determine significant differences between means at α = 0.05.

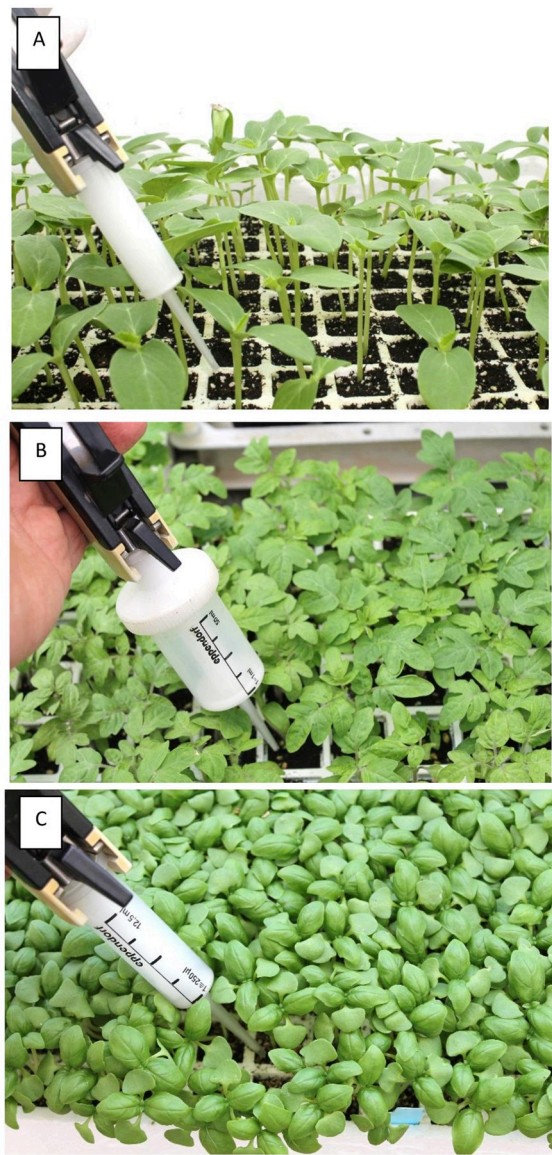

**Fig 1. Application of fungicides to the root system of young cucumber, tomato and basil plants growing in multi-cell trays.** One milliliter of a fungicide suspension is applied to the soil surface of each cell (cell size 5x5 cm for cucumber and tomato; 2.5x2.5 cm for basil) with the aid of a 12-ml or 50-ml Eppendorf dispenser. At 1h and 5h after root treatment, 2 ml of water were added to the soil surface of each treated plant to facilitate the uptake of the fungicide by the root system. **A**- Cucumber. **B**- Tomato. **C**- Basil.

Six field experiments were performed with cucumber and six field experiments with basil, with one or two replicate plots (0.6 x 1.2m each) per treatment and 8–10 plants per plot. Percent season-long protection for each fungicide was calculated as the ratio between the AUDPC (area under disease progress curve) of the treated plants and the AUDPC of the control fungicide-free plants [4, 29].

The synergistic interaction between oxathiapiprolin and benthiavalicarb was calculated according to Abbot or Wadley as described before [30, 31].

## Results

### Growth chamber experiments

**Tomato late blight.** **Exp. 1.** Six leaf tomato plants (*cv* Hazera 129) grown in 250 ml pots were soil-drenched with 0.5 or 1 mg a.i. of BENT, OXPT or ZE. Plants were inoculated a day later with *P. infestans* and assessed for disease development at 8 dpi (days post inoculation). MIC (minimal inhibitory concentration) values for BENT, OXPT and ZE were 1, > 1 and 0.5 mg a.i./plant, respectively (Fig 2A). ZE at 0.5 mg ai/plant provided a significantly reduced blight than the other fungicides at the same dose thus exhibiting a synergy factor (SF) of 1.2 (Abbott). The appearance of the plants at 8 dpi is shown in Fig 2B–2D.

**Exp. 2.** ZE is a mixture composed of OXPT and BENT at a ratio of 3+7 (w/w). The following experiment was performed to learn if synergistic interaction occurs between the components of this mixture. The root system of 6-leaf tomato plants was treated with BENT of 0.007, 0.07, 0.7; OXPT of 0.003, 0.03, 0.3, or ZE of 0.01, 0.1, 1 mg a.i/ plant. Plants were inoculated with *P. infestans* at 2 days after treatment and scored for late blight symptoms at 8 dpi. The results are given in Fig 3. Increased doses of OXPT, BENT or ZE increased disease control (Fig 3A and 3B). ZE was significantly the most effective fungicide. Probit analysis revealed ED90 values of 0.222, 0.164 and 0.04 mg a.i. for BENT, OXPT and ZE, respectively. Based on these values the expected ED90 of ZE is 0.200, thus providing a high SF value (Wadley) of 5.

**Exp. 3**. Five leaf tomato plants grown in 250 ml pots (n = 40) were soil-drenched at 0.3, 0.7 or 1 mg a.i. of OXPT, BENT and ZE, respectively. They were inoculated with *P. infestans* at 3 DAT (days after transplanting) and scored at 7 dpi. The results are presented in Fig 4A–4E. The data show a significantly better control of late blight by ZE compared to BENT or OXPT (Fig 4A). Percent protection provided by OXPT, BENT and ZE was 87, 93 and 100%, respectively.

**Exp. 4**. The long-term protection against late blight by root treatment with ZE was examined as follows: 3-leaf tomato plants growing in multi-cell trays were treated by a soil drench with ZE of 0.5 or 1 mg a.i per plant. They were transferred to 250 ml pots a day later and were taken for inoculation (n = 10) at various time intervals after treatment. The results presented in Fig 5 show complete control of the disease up to 42 DAT. At 54 DAT, the lower dose of ZE was partially effective, while the higher dose was fully effective. At 81 DAT, when plants developed 20 leaves, ZE of 0.5 and 1 mg a.i. provided 80 and 85% protection, respectively (Fig 5A and 5B).

**Cucumber downy mildew.** **Exp. 1**. The root system of 3-leaf cucumber plants grown in 250 ml pots were treated with 1 mg a.i. of BENT, OXPT or ZE. Plants were inoculated with sporangia of *P. cubensis* at the 5-leaf stage and scored for disease symptoms at 14 dpi. The results presented in Fig 6A show that BENT provided partial, though significant, protection compared with untreated plants, whereas OXPT and ZE provided complete control of the disease.

**Exp. 2**. One-leaf cucumber plants grown in multi-cell trays were soil drenched (n = 12) with 0.01, 0.1 or 1 mg a.i. of BENT, OXPT or ZE. Plants were inoculated with *P. cubensis* at 3 DAT. Disease recorded at 7 dpi are shown in Fig 6B. BENT, OXPT and ZE of 0.01 mg a.i./ plant provided 39, 75 and 100% protection against the disease, respectively with SF (Abbott) of 1.18. The results showed a significantly higher efficacy of ZE compared to OXPT, and of OXPT compared to BENT.

**Basil downy mildew.** **Exp. 1**. Twelve-leaf basil plants grown in 250 ml pots were treated (n = 6) with 1 mg a.i. of OXPT or ZE, inoculated with sporangia of *P. belbahrii* two days later and evaluated for disease development at 12 dpi. Percent sporulating leaf area in control

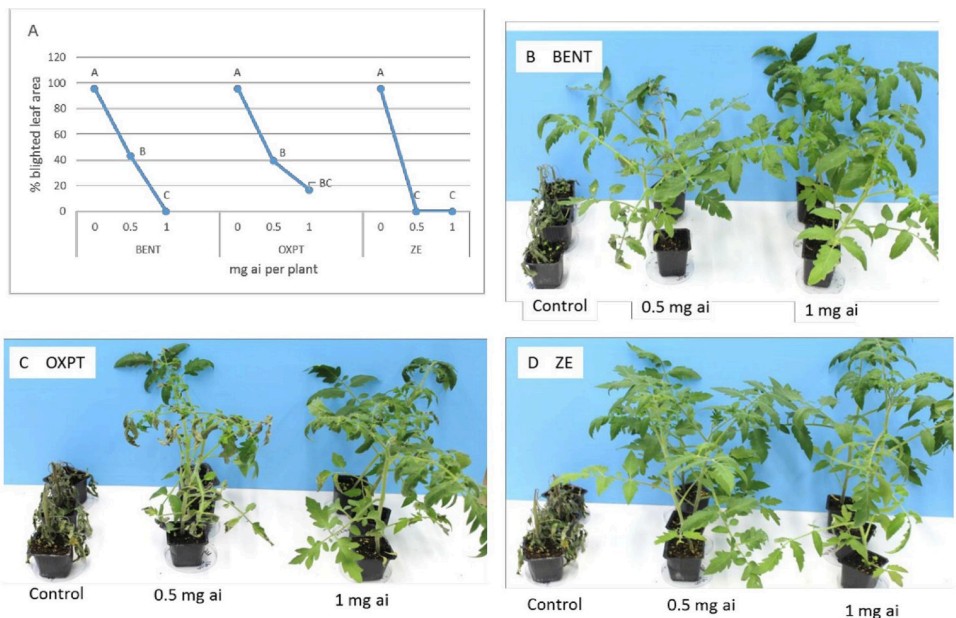

**Fig 2. Control of late blight by BENT, OXPT and ZE applied to the root system of tomato plants.** Plants (n = 4) were inoculated with sporangia of *P. infestans* at 8h after root treatment and scored for disease development at 7 dpi. **A**- Percentage blighted leaf area. Different letters on curves indicate on a significant difference at α = 0.05 (t-test). **B**-ED90 values derived from the values in A after log-probit transformation. ZE provides SF value of 22.2 (Wadley). **C-E**-the appearance of the inoculated plants at 7 dpi.

plants was 68.3±7.5. OXPT provided 59% control of the disease, whereas ZE provided 100% control of the disease.

**Exp. 2**. Two-leaf basil plants grown in multi-cell trays were treated (n = 20) with ZE of 0.25, 0.5 or 1 mg a.i. per plant applied to the root system. They were transferred to 250 ml pots and inoculated at 7 DAT. Percent leaf area sporulating at 11 dpi in control-inoculated plants was 59.8±18.4 percentage, whereas none of the treated plants showed any sporulation of *P. belbahrii*, regardless of the ZE dose applied.

## Field experiments

**Cucumber downy mildew.   Exp. 1.** The root system of 2-leaf cucumber plants growing in 250 ml pots was treated (n = 40) with ZE of 0.01, 0.1 or 1 mg a.i. per plant. Untreated plants served as control. Plants were transplanted to the field at 1DAT (14.2.2019) and scored for downy mildew development for 4 weeks. Disease progress with time is shown in Fig 7A and AUDPC in Fig 7B. Season-long protection provided by ZE of 0.01, 0.1 and 1 mg a.i./plant was 26, 53 and 100%, respectively (Fig 7C).

**Exp. 2**. The root system of 1-leaf cucumber plants grown in 250 ml pots was treated with 1mg a.i. per plant of BENT, OXPT or ZE and transplanted to the field at 13 DAT (28.4.2019). Downy mildew lesions appeared in control plots at 40 DAT, as against 52 DAT in BENT-treated plots. No disease was seen in OXPT or ZE treated plots until 54 DAT (Fig 7D and 7E). Season-long protection provided by BENT, OXPT and ZE was 84, 100 and 100%, respectively (Fig 7F). Minor disease symptoms developed in ZE-treated plots at 56 DAT.

**Exp. 3**. The root system of 1-leaf cucumber plants growing in multi-cell trays were treated with 1 mg a.i. per plant of OXPT, BENT or ZE and transplanted to the field one day later (12.5.2019). First lesions of downy mildew were seen in control plots at 15 DAT. Disease

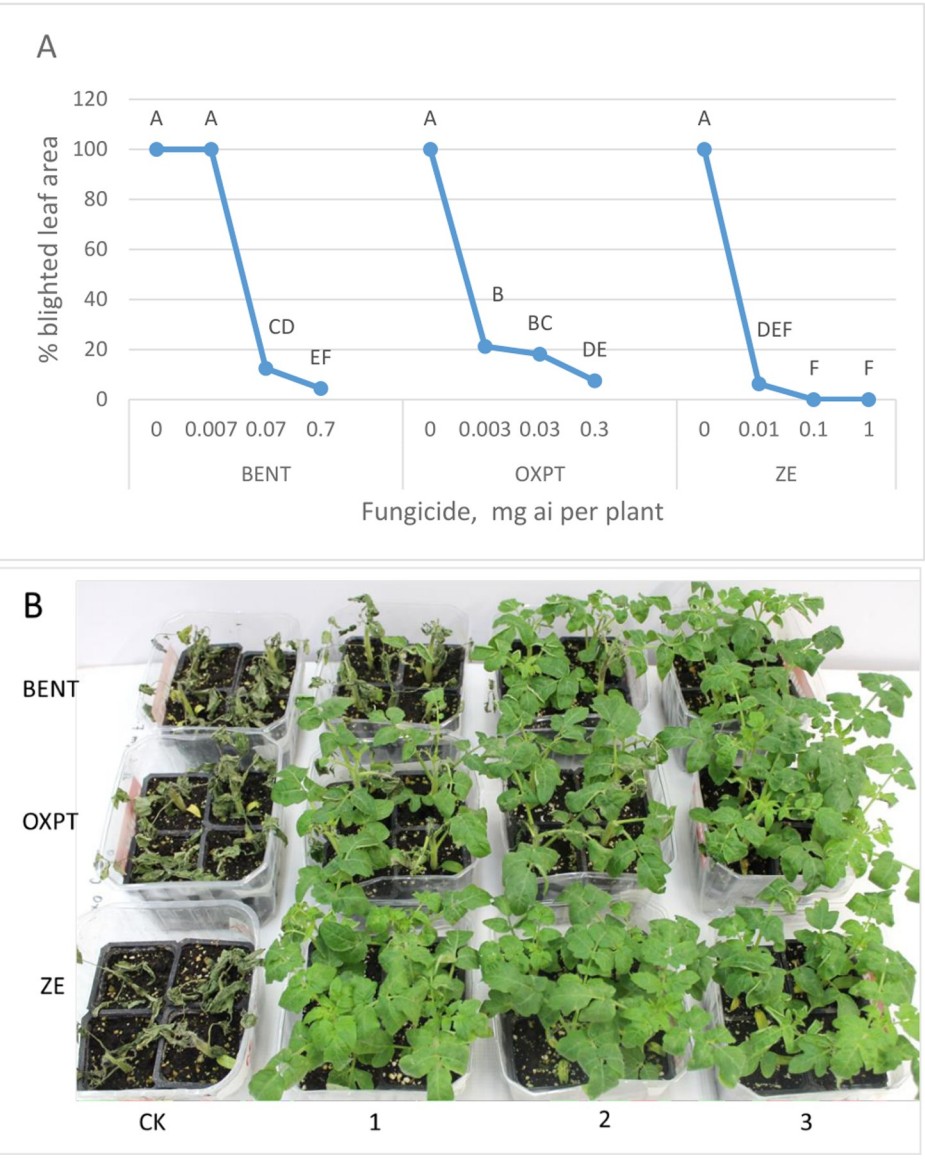

**Fig 3. Control of late blight in tomato by root treatment with fungicides. A**- Percentage of blighted leaf area at 8 dpi. Different letters on bars indicate on a significant difference between treatments at $\alpha$ = 0.05 (t-test). **B**- The appearance of the inoculated at 8 dpi. CK = Control untreated. 1, 2, 3 = the dose of a fungicide applied to the root system of each plant: BENT- 0.007, 0.07 0.7; OXPT- 0.003, 0.03, 0.3; ZE- 0.01, 0.1, 1 mg a.i. per plant. Plants were inoculated with sporangia of *P. infestans* at 2 days after treatment.

progress with time is shown in Fig 8A and AUDPC in Fig 8B. By the end of the experiment, the plants had 12–14 leaves and carried a mean of 1350±212 lesions per plant. All three fungicides suppressed disease development in a significant manner (Fig 8A). BENT, OXPT and ZE provided 89, 98 and 99% long season protection, respectively (Fig 8C).

**Exp. 4.** The root system of 1-leaf cucumber plants grown in multi-cell trays were treated with 1mg a.i. per plant of OXPT, BENT or ZE and transplanted to the field four days later (11.6.2019). Disease progress with time is shown in Fig 8D, and AUDPC and season-long protection values are presented in Fig 8E and 8F, respectively. Final disease score in the control

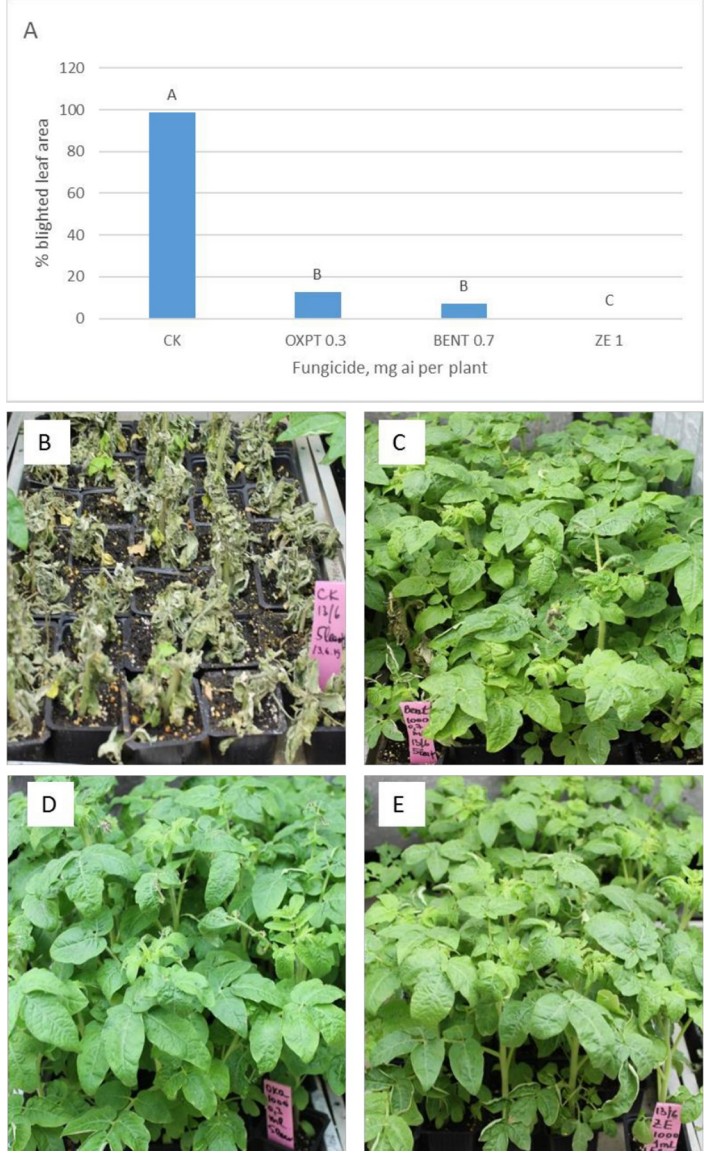

**Fig 4. Control of late blight in tomato by root treatment with fungicides.** Plants (n = 40) were inoculated with *P. infestans* at three DAT and photographed at 7dpi. **A**- Percentage blighted leaf area at 7dpi. Different letters on bars indicate on a significant difference between treatments at α = 0.05 (t-test). **B**- Control untreated inoculated plants. **C**- BENT 0.7 mg a.i. /plant. **D**- OXPT- 0.3 mg a.i./plant. **E**- ZE- 1 mg a.i./plant.

plots was significantly higher than in the treated plots (40, 5 and 10% in BENT, OXPT, and ZE treated plots, respectively) (Fig 8D). Final disease scores in OXPT and ZE treated plots were significantly lower than in BENT treated plots, but did not differ significantly among themselves. ZE was synergistically effective, with SF values (Abbott) of 1, 1, 1.03, 1.02, 1.08 and 1.06 at 10, 13, 15, 17, 20 and 22 DAT, respectively.

**Exp. 5.** Cucumber plants at their 1-leaf stage were soil-drench with 1ml suspension containing 0.01, 0.1 or 1 mg a.i. of OXPT, BENT or ZE (n = 10). Plant were transplanted to the field after one day (11.7.2019) and disease progress was followed for 24 days. The results are presented in Fig 9A–9C. First disease lesions appeared at 15 DAT with a mean of 100 lesions colonizing each control plant. Within 9 days, a mean of 800 lesions developed on each control

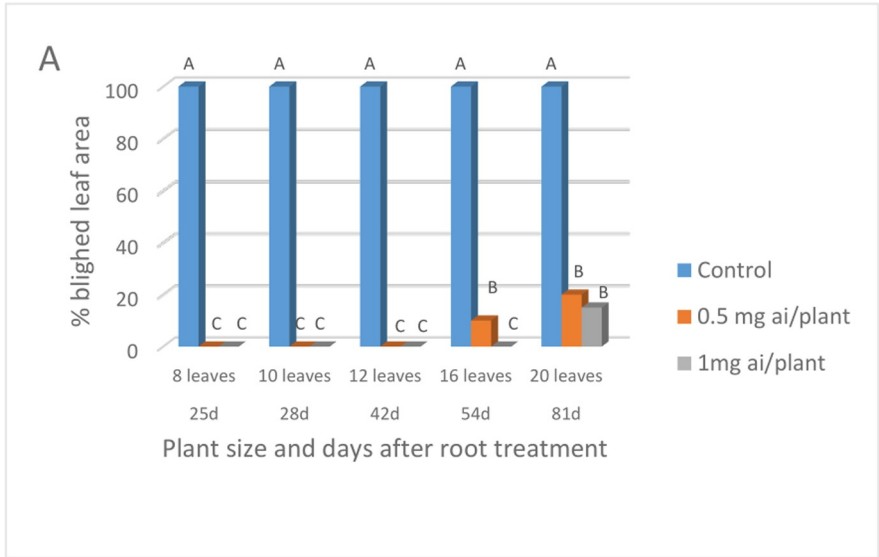

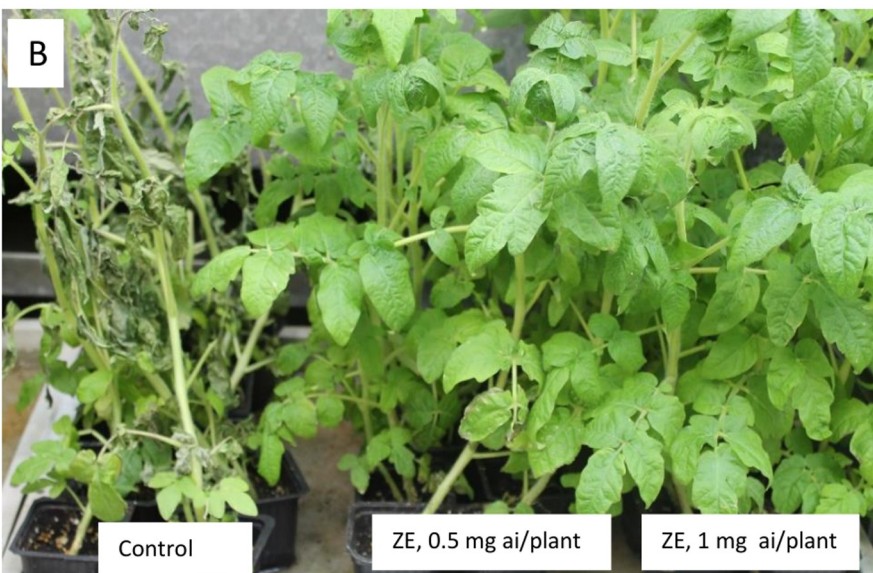

**Fig 5. Long-term protection of tomato against late blight by ZE applied to the root system. A**- Percentage blighted leaf area in plants taken for inoculation with *P. infestans* at various time intervals after soil drench with ZE. Different letters on bars indicate on a significant difference at $\alpha = 0.05$ (n = 10, t-test). **B**- The appearance of the tomato plants taken for inoculation at 42 days after treatment. Photo taken at 7 dpi.

plant, which bears 12 leaves (Fig 9A). All fungicides suppressed disease development in a significant manner regardless of the dose used (except OXPT of 0.01 mg ai/plant) (Fig 9A). AUDPC values are shown in Fig 9B. Season-long protection provided by OXPT, BENT and ZE ranged between 59–100, 83–97 and 81–95%, respectively (Fig 9C).

**Exp.6**. Cucumber plants at their 1-leaf stage were soil-drench with 1ml suspension containing 0.01, 0.1 or 1 mg a.i. of OXPT, BENT or ZE (n = 15). Plant were transplanted to the field at 4 DAT (1.8.2019). Disease records were taken from 12 to 23 DAT. All fungicides suppressed disease development in a significant manner regardless of the dose used (Fig 9D). AUDPC and season-long protection data are shown in Fig 9E and 9F, respectively. The appearance of the

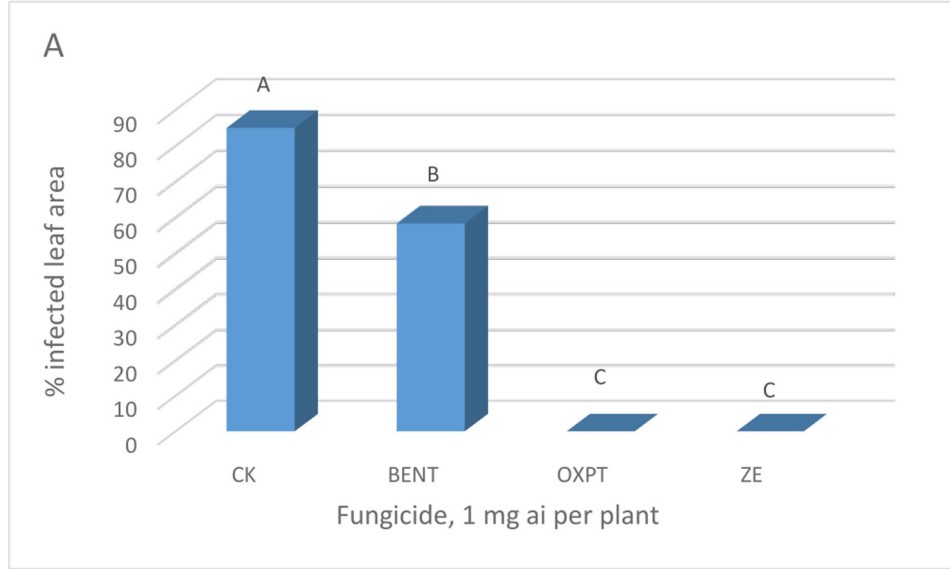

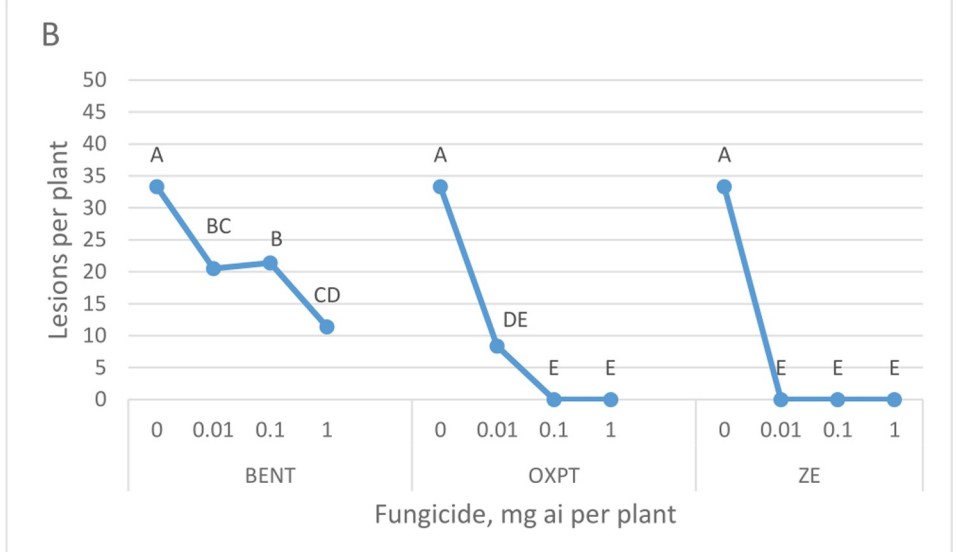

**Fig 6. Control of downy mildew in cucumber by fungicides applied to the root system. A**- The root system of 3-leaf cucumber plants growing in 250 ml pots were treated with 1 mg a.i. of BENT, OXPT or ZE. Plants (n = 8) were inoculated at the five leaf stage with sporangia of *P. cubensis* and scored for disease at 14 dpi. Different letters on bars indicate on a significant difference between treatments at α = 0.05 (t-test). **B**- Protection of cucumber plants (n = 12) against downy mildew by root treatment with BENT, OXPT and ZE. Different letters on bars indicate on a significant difference between treatments at α = 0.05 (t-test).

plants at the end of the season is shown in Fig 9H and 9I. Based on the data in presented in Fig 9E a synergy factor (Wadley) of 8.03 for ZE was calculated.

**Basil downy mildew. Exp. 1.** The root system of 2-leaf basil plants grown in multi-cell trays were treated with OXPT of 0.003, 0.03 or 0.3, BENT of 0.007, 0.07 or 0.7, or ZE of 0.01, 0.1 or 1 mg a.i. per plant. Plants were transplanted to the field at 1 DAT (5.6.2019). Disease records as taken at 12, 15 and 20 DAT are shown in Fig 10A. OXPT and ZE were highly effective in protecting the plants from downy mildew, whereas BENT was partially effective (Fig 10A). AUDPC values (Fig 10B) and season-long protection values (Fig 10C) indicated on higher

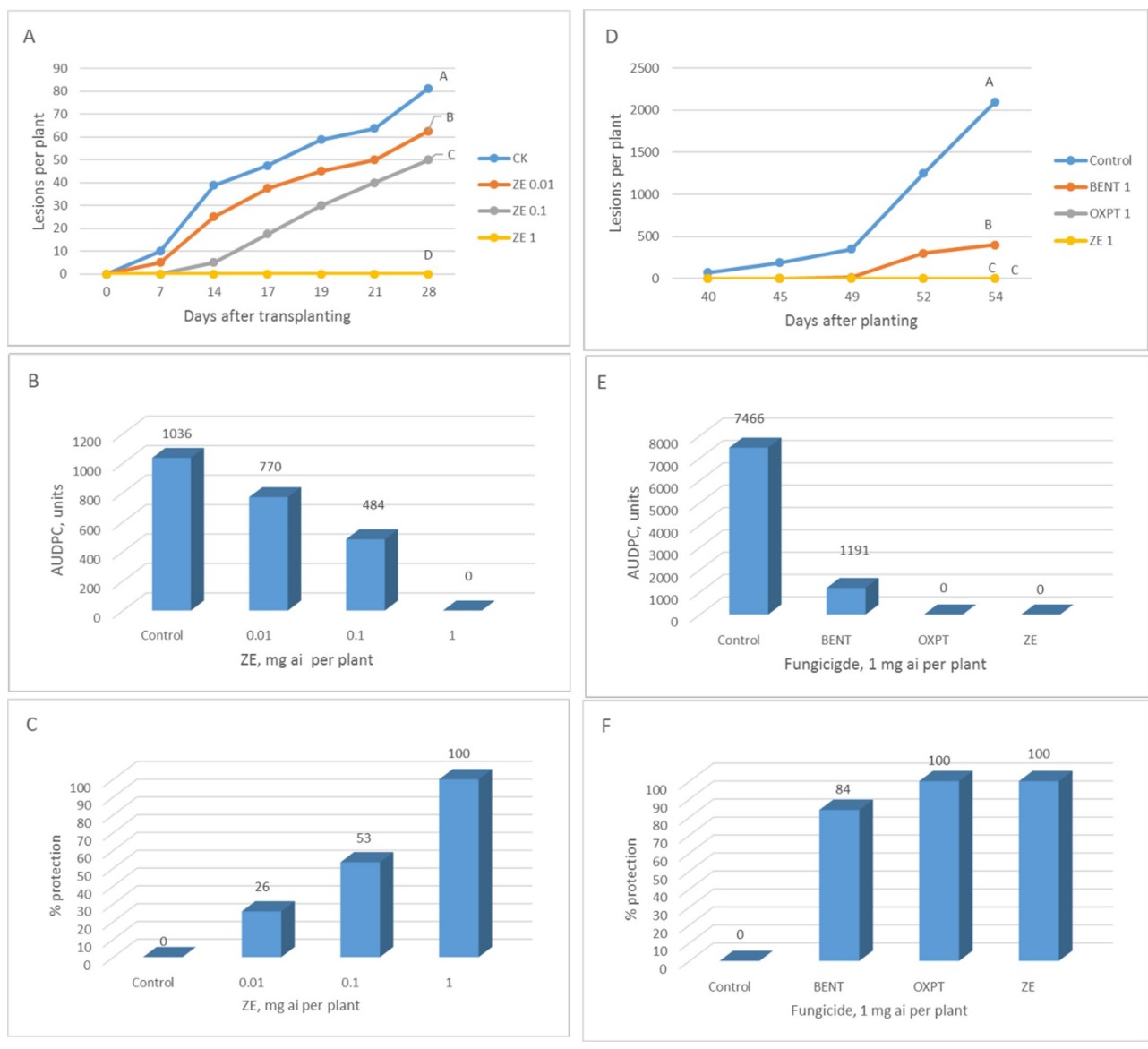

**Fig 7. A-C,** Protection of cucumber plants in the field against downy mildew by root treatment with ZE. Two leaf plants growing in 250 ml pots were treated (n = 40) by ZE of 0.01, 0.1 or 1 mg a.i. per plant and transplanted to the field one day later. **A**- Disease progress with time. Different letters on curves indicate on a significant difference between treatments (α = 0.05) at the end of the experiment (t-test). **B**- Area under disease progress curve. **C**- Season-long protection. **D-F,** Control of cucumber downy mildew in the field in by root treatment with fungicides. One-leaf cucumber plants growing in 250 ml pots were treated by soil drench with 1 mg a.i. of BENT, OXPT or ZE and transplanted to the field at 13 DAT. **D**- Number of lesions developed per plant at various time intervals after inoculation. Different letters on curves indicate on a significant difference between treatments (α = 0.05) at the end of the experiment (t-test). **E**- Area under disease progress curve. **F**- Season-long protection.

efficacy of ZE compared to OXPT or BENT. ED90 values at 20 DAT for OXPT, BENT and ZE were 0.123, >0.700 and 0.645 mg a.i. per plant, respectively, with ZE providing a synergy factor (Wadley) of 3.28.

**Exp. 2**. The root system of 2-leaf basil plants growing in multi-cell trays were treated with 0.25, 0.5 or 1 mg a.i. of OXPT, BENT or ZE and transplanted to the field at 10 DAT (16.5.2019). Disease progress curves, AUDPC and season-long protection values are shown in Fig 11A–11C. The disease in control untreated plants was seen at 17 DAT as against 26 DAT in root-treated plants (Fig 11A). Disease level in control plants reached 87.5% at 26 DAT.

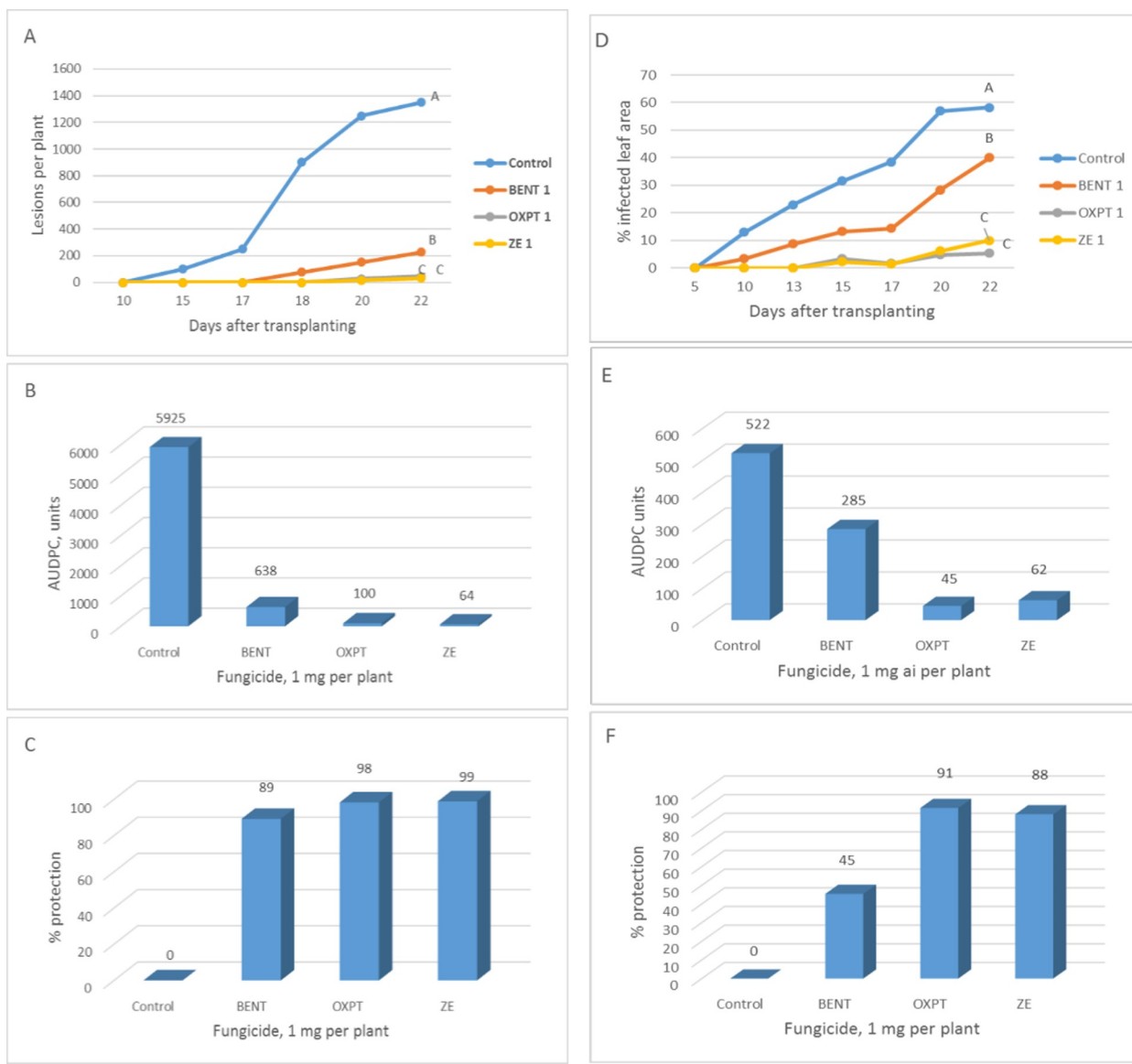

**Fig 8. A-C,** Control of cucumber downy mildew in the field in by root treatment with BENT, OXPT or ZE. One-leaf plants growing in multi-cell trays were treated by a soil drench with 1 mg a.i. of a fungicide and transplanted to the field one day later. **A**- Disease progress with time. Different letters on curves indicate on a significant difference between treatments at the end of the experiment ($\alpha$ = 0.05). **B**- Area under disease progress curves. **C**- Season-long protection. **D-F,** Control of cucumber downy mildew in the field in by root treatment with BENT, OXPT or ZE. One-leaf plants growing in multi-cell trays were treated by a soil drench with 1 mg a.i. of a fungicide and transplanted to the field four days later. **D**- Disease progress with time. Different letters on curves indicate on a significant difference between treatments at the end of the experiment ($\alpha$ = 0.05). **E**-Area under disease progress curves. **F**- Season-long protection.

AUDPC and season-long protection are shown in Fig 11B and 11C, respectively. The most and least effective treatments were OXPT 1 mg a.i. per plant and BENT 0.25 mg a.i. per plant, respectively. The observed ED90 values of ZE at 26, 30 and 33 DAT were 0.215, 0.509 and 2.874 mg a.i./plants, respectively, whereas the expected ED90 values of ZE (Wadley) were 0.417, 1 and 3.333 mg a.i./plants, respectively (Fig 11D). Thus, the observed efficacy of ZE at 26, 30 and 33 DAT was x1.94, x1.96 and x1.15 higher (synergy factor) than expected (Fig 11E). The appearance of the plants at 28 DAT is shown in Fig 11F and 11G. Control plants (Fig 11F)

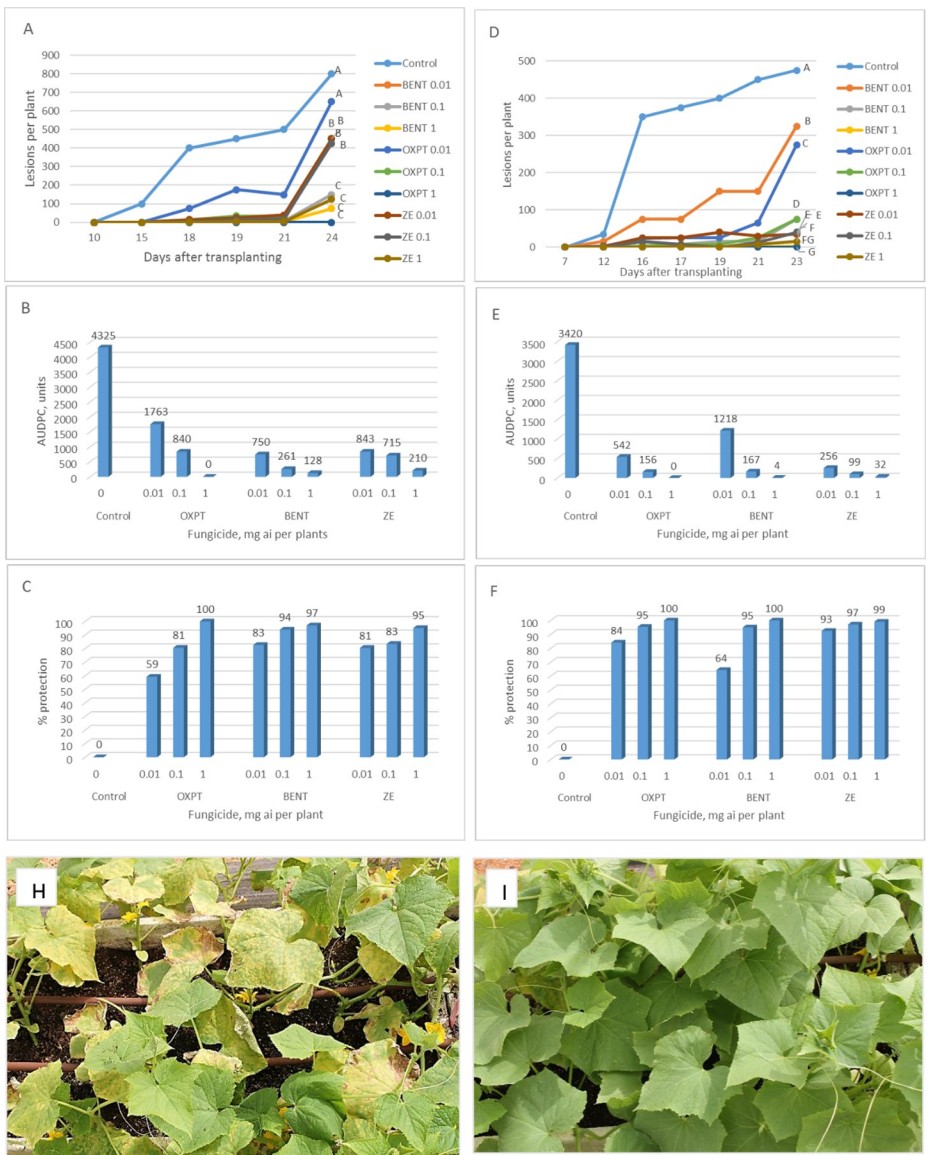

**Fig 9. A-C,** Control of cucumber downy mildew in the field in by root treatment with BENT, OXPT or ZE. One-leaf cucumber plants (n = 10) growing in multi-cell trays were treated with BENT, OXPT or ZE of 0.01, 0.1 or 1 mg a.i. per plant. Plants were transplanted to the field one day later. Disease records were taken at 10–24 DAT. **A**- Disease records as taken at various time intervals after transplantation. Different letters on curves indicate on a significant difference between treatments at $\alpha = 0.05$ (t-test). **B**- Area under disease progress curves. **C**- Season-long protection. **D-F,** Control of cucumber downy mildew in the field in by root treatment with BENT, OXPT or ZE. One-leaf cucumber plants (n = 10) growing in multi-cell trays were treated with BENT, OXPT or ZE of 0.01, 0.1 or 1 mg a.i. per plant. Plants were transplanted to the field three days later. Disease records were taken at 7–23 DAT. **D**- Disease records as taken at various time intervals after transplantation. Different letters on curves indicate on a significant difference between treatments at $\alpha = 0.05$ (t-test). **E**- Area under disease progress curves. **F**- Season-long protection. **H**- The appearance of control untreated plants. Note the multiple number of downy mildew lesions. **I**- The appearance of plants treated with OXPT of 1 mg ai per plant. Note the absence of mildew lesions. Photos taken at 23 DAT.

were devastated of downy mildew whereas plants treated with 1 mg ai of OXPT were completely healthy (Fig 11G).

**Exp. 3.** The root system of 2-leaf basil plants growing in multi-cell trays were treated (n = 8) with 1 mg a.i. of OXPT, BENT or ZE. Untreated plants served as control. After one day, the plants were planted in 250 ml pots, and 10 days later (20.6.2019), they were transplanted to the

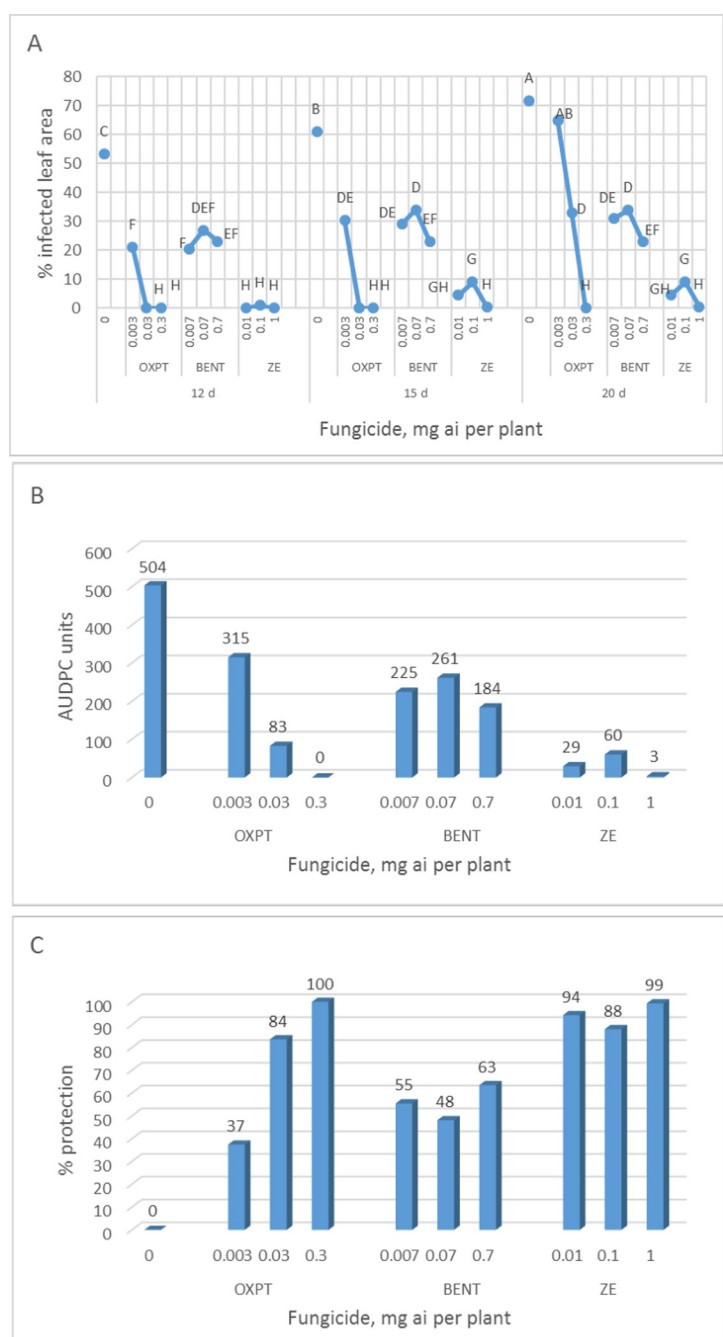

**Fig 10. Control of basil downy mildew in the field by root treatment with BENT, OXPT or ZE.** Two-leaf basil plants (n = 10) growing in multi-cell trays were treated with OXPT of 0.003, 0.03 or 0.3, BENT of 0.007, 0.07 or 0.7 and ZE of 0.01, 0.1 or 1 μg a.i. per plant. Plants were transplanted to the field one day later. **A**- Disease records as taken at various time intervals after transplantation. Different letters on curves indicate on a significant difference between treatments at α = 0.05 (t-test). **B**- Area under disease progress curves. **C**- Season-long protection.

field. Disease records were taken from 7 to 24 DAT. Disease progressed very fast, reaching 92.5 ±2.7% infected leaf area in control plants at 24 DAT. All fungicides suppressed the disease in a significant manner reaching 2.5±7.1, 2.5±4.6 and 5.6± 6.2% infected leaf area in plants treated with BENT, OXPT and ZE, respectively (Fig 12A).

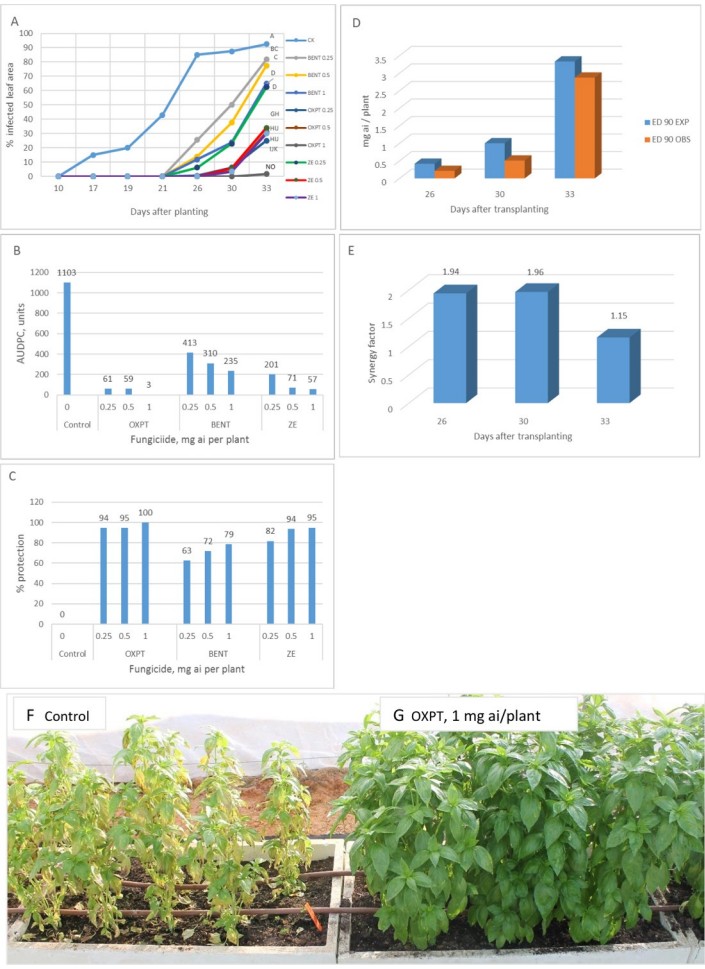

**Fig 11. Control of basil downy mildew in the field by root treatment with BENT, OXPT or ZE.** Two-leaf basil plants growing in multi-cell trays were treated with BENT, OXPT or ZE of 0.25, 0.5 or 1 mg a.i. per plant. Plants were transplanted at one DAT to 250 ml pots and at 10 DAT to the field. **A**- Disease records as taken at various time intervals after transplantation. Different letters on curves indicate on a significant difference between treatments at α = 0.05 (t-test). **B**- Area under disease progress curves. **C**- Season-long protection. **D**- Observed and expected ED 90m values of ZE. **E**- Synergy factor for ZE. **F**- The appearance of control untreated plants. Note intense yellowing due to heavy infection with downy mildew. **G**- The appearance of plants treated with OXPT of 1 mg a.i./plant. Note the intense green foliage free of infection with downy mildew. Photos taken at 28 DAT.

**Exp. 4.** A similar experiment conduced a month later (19.7.2019) provided similar results (Fig 12B).

**Exp. 5.** The root system of 2-leaf basil plants growing in multi-cell trays were treated with 0.01, 0.1 or 1 mg a.i. of OXPT, BENT or ZE (n = 10) and transplanted to the field one day later (22.7.2019). The results are presented in Fig 13A–13C. The disease showed up in control plants at 9 DAT and progressed rapidly, reaching 85% infected leaf area at 23 DAT (Fig 13A). A negative association was recorded between the dose of a fungicide applied, disease intensity and season-long protection (Fig 13B and 13C). Root treatment at 1 mg ai of either fungicide provided 98% protection during the 23 days of the epidemics (Fig 13C).

**Exp. 6.** The root system of 4-leaf basil plants growing in 250 ml pots were treated with 0.01, 0.1 or 1 mg a.i. of OXPT, BENT or ZE (n = 10) and transplanted to the field one day later (29.7.2019). The results are presented in Fig 13D–13F. The disease appeared at 8 DAT

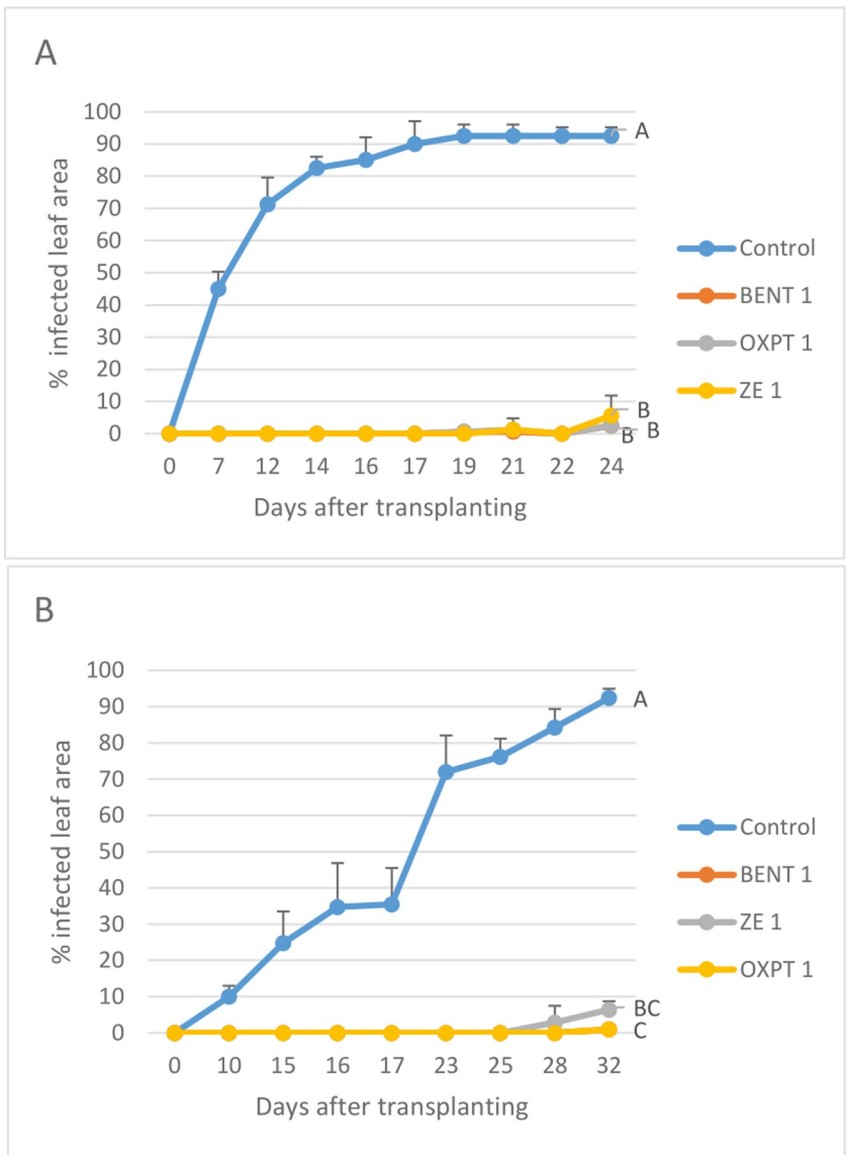

**Fig 12. Two experiments showing the control of basil downy mildew in the field by root treatment with 1 mg ai per plants of BENT, OXPT or ZE. A**- Six-leaf plants growing in 250 ml pots (n = 8) were treated with a fungicide and transplanted to the field at 10 days after treatment. Disease scores were taken at 7–24 DAT. Different letters on curves indicate on a significant difference between treatments at α = 0.05 (t-test). **B**- Two-leaf basil plants growing in multi-cell trays were treated (n = 20) with a fungicide and transplanted to the field at two days after treatment. Disease scores were taken at 10–32 DAT. Different letters on curves indicate on a significant difference between treatments at α = 0.05 (t-test).

attacking 85% of the leaves in control plants at the end of the epidemic. All fungicides at 0.1 and 1 mg per plants suppressed the disease in a significant manner. A clear positive association occurred between fungicide dose and disease inhibition. The higher the dose was the stronger was disease suppression (Fig 13D and 13E) and higher was the protection achieved (Fig 13F). The appearance of the plants at 13 DAT is shown in Fig 13H and 13I. Profuse sporulation of *P. belbahrii* was observed on lower leaf surfaces of control plants (Fig 13H), while no sporulation was seen at this stage in plants treated with ZE of 0.01 mg ai/plant (Fig 13I).

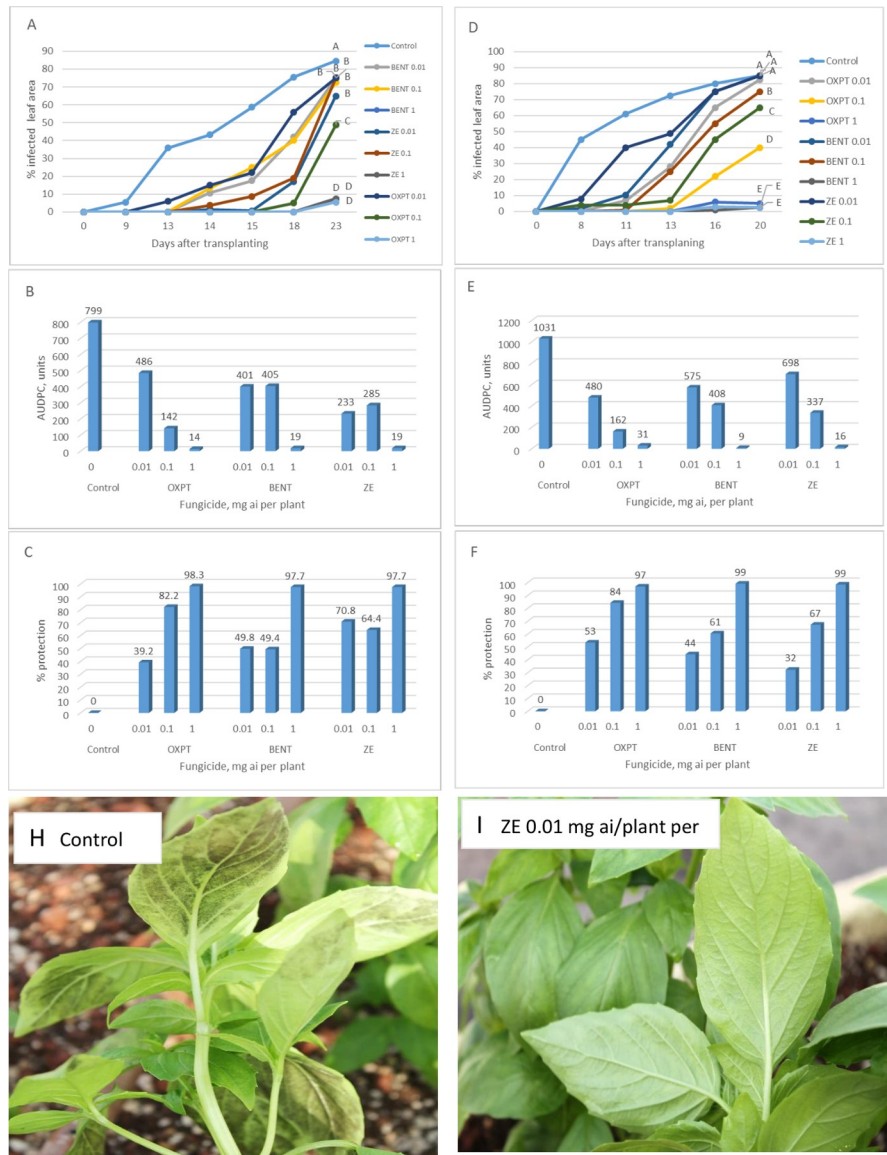

**Fig 13.  A-C,** Control of basil downy mildew in the field by root treatment with BENT, OXPT or ZE. Two-leaf basil plants growing in multi-cell trays were treated (n = 10) with BENT, OXPT or ZE of 0.01, 0.1 or 1 mg a.i. per plant. Plants were transplanted to the field one day later. Disease scores were taken at 9–23 DAT. **A-** Disease records as taken at various time intervals after transplantation. Different letters on curves indicate on a significant difference between treatments at α = 0.05 (t-test). **B-** Area under disease progress curves. **C-** Season-long protection. **D-F,** Control of basil downy mildew in the field by root treatment with BENT, OXPT or ZE. Four-leaf plants growing in 250 ml pots (n = 10) were treated with a fungicide and transplanted to the field two days after treatment. Disease scores were taken at 8–20 DAT. **D-** Disease records as taken at various time intervals after transplantation. Different letters on curves indicate on a significant difference between treatments at α = 0.05 (t-test). **E-** Area under disease progress curves. **F-** Season-long protection. **H** and **I**, the appearance of the plants at 13 DAT. Note heavy sporulation of *P. belbahrii* in control plants but no sporulation in plants treated with ZE of 0.01 mg ai per plant.

## Discussion

In previous studies, we showed that oxathiapiprolin (OXPT) is effective against oomycete foliar pathogens when applied to the root system of cucumber and tomato plants [3, 4, 9]. OXPT exhibits systemic activity in spite of its low solubility in water (0.175 mg/L at 20˚C) and

high octanol/water partition coefficient (Log Pow 3.62–3.67 at pH 4–9). The extremely high intrinsic activity of this compound, with ED90 values of about 1ppb a.i. against spore germination [3], implies that even a minor quantity of OXPT taken up by the root system is sufficient to suppress disease development in the foliage.

According to FRAC (http://www.frac.info/), resistance risk of OXTP (FRAC code U15) is assumed medium to high and resistance management is required. To avoid the buildup of fungal resistance, the producers mix it with other fungicides, which have a different mode of action. We tested four OXPT mixtures (with chlorothalonil, azoxystrobin, mandipropamid or mefenoxam) in root treatments against downy mildew in cucumber and late blight in tomato. The most effective mixture was OXPT+mefenoxam [4, 9]. The enhanced efficacy of this mixture was attributed to the systemic nature of mefenoxam and its formulation.

Another systemic anti-oomycete fungicide is benthiavalicarb-isopropyl (BENT) (solubility in water 13.14 mg/L at 20˚C and partition coefficient Log Pow 2.52 at 25˚C, [32]). A mixture of oxathiapiprolin+benthiavalicarb-isopropyl (30g ai+70g ai per Kg) is now available as Zorvec Endavia100 OD (= ZE).

It is shown here that ZE and its components OXPT and BENT provide effective systemic protection against late blight in tomato and downy mildews in cucumber and basil following its application to the root system. In growth chamber experiments, a single root treatment with 1 mg a.i. of either fungicide provided complete control of late blight in tomato, downy mildew in cucumber and downy mildew in basil. Full protection in tomato persisted for as long as 42 days. In field experiments, a single application to the root system at 1 mg a.i. of either compounds provides a season-long protection against cucumber and basil downy mildews. Often, ZE performed better than OXPT and BENT combined, suggesting a synergistic interaction between the two components.

Authorizations exist for the use of OXPT as foliar treatments or as soil treatments at planting (band/in-furrow or in transplant water) or via drip irrigation in North America, Canada, a number of countries in the Pacific, Asia, Central and South Americas and are pending in Europe [27]. However, no information is available on the efficacy of root treatments with OXPT or its mixtures against foliar oomycete diseases of vegetables or herbs.

A major technical finding of the present study was that root treatment given to young seedlings growing in multi-cell trays in the nursery is an effective disease control measure. A single root treatment given to 2–3 leaf plants growing in multi-cell trays was effective upon transplanting against late blight in growth chamber until the 20-leaf stage in tomato, and in the field, against natural infection until the ~20-leaf stage in cucumber and ~ 100-leaf stage in basil.

The label of ZE recommends spray of 40-gram a.i. per hectare. With a density of 25,000 cucumber plants per hectare, this provides a dose of 1.6 mg a.i. per plant. When spray is repeated after about two weeks, this dose is doubled to 3.2 mg a.i. per plant. The new technique described here provides 3-4-week protection with only 1 mg a.i. of ZE per plant, thus reducing the dose applied by about 70%. Obviously, such decrease in the dose applied shall also reduce the risk of resistance buildup.

The three foliar oomycete pathogens studied here are all sensitive to OXPT and BENT. Both compounds are systemic. The mixture ZE provides simultaneous translocation of both fungicides to the leaf tissue of plants treated via the root system. ZE, therefore, provides not only excellent protection against disease but also mutual protection against developing resistance towards either OXPT or BENT.

## Acknowledgments

I acknowledge Dr. M. Galperin for her assistance in statistical analyses of the data.

## Author Contributions

**Conceptualization:** Yigal Cohen.

**Data curation:** Yigal Cohen.

**Formal analysis:** Yigal Cohen.

**Investigation:** Yigal Cohen.

**Methodology:** Yigal Cohen.

**Project administration:** Yigal Cohen.

**Validation:** Yigal Cohen.

**Writing – original draft:** Yigal Cohen.

**Writing – review & editing:** Yigal Cohen.

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
