## [Decision Letter · Decision Letter 0]

3 Oct 2019

PONE-D-19-24369

Root treatment with oxathiapiprolin, benthiavalivarb or their mixture provides prolonged systemic protection against oomycete foliar pathogens

PLOS ONE

Dear Prof. Cohen,

Thank you for submitting your manuscript to PLOS ONE. After careful consideration, we feel that it has merit but does not fully meet PLOS ONE’s publication criteria as it currently stands. Therefore, we invite you to submit a revised version of the manuscript that addresses the points raised during the review process.

We would appreciate receiving your revised manuscript by Nov 17 2019 11:59PM. To enhance the reproducibility of your results, we recommend that if applicable you deposit your laboratory protocols in protocols.io, where a protocol can be assigned its own identifier (DOI) such that it can be cited independently in the future. For instructions see: http://journals.plos.org/plosone/s/submission-guidelines#loc-laboratory-protocols

We look forward to receiving your revised manuscript.

Kind regards,

Binod Bihari Sahu, Ph.D.

Academic Editor

PLOS ONE

Journal Requirements:

1. We noticed you still have some minor occurrence of overlapping text with the following previous publication(s), which needs to be addressed:

https://link.springer.com/article/10.1007%2Fs10658-010-9698-6

In your revision ensure you cite all your sources (including your own works), and quote or rephrase any duplicated text outside the methods section. Further consideration is dependent on these concerns being addressed.

Additional Editor Comments (if provided):

Please address to the comments and rectify before re-submission.

Reviewers' comments:

Reviewer's Responses to Questions

**Comments to the Author**

1. Is the manuscript technically sound, and do the data support the conclusions?

Reviewer #1: Yes

Reviewer #2: Yes

2. Has the statistical analysis been performed appropriately and rigorously? 

Reviewer #1: Yes

Reviewer #2: Yes

3. Have the authors made all data underlying the findings in their manuscript fully available?

Reviewer #1: Yes

Reviewer #2: Yes

4. Is the manuscript presented in an intelligible fashion and written in standard English?

Reviewer #1: No

Reviewer #2: Yes

5. Review Comments to the Author

Reviewer #1: While the scientific method performed in the paper is sound, I believe significant editing for clarity/cohesiveness needs to be done before the paper is accepted.

There are many small paragraphs that I believe could be put together to form a more coherent narrative.

When referring to a disease (i.e. downy mildew of basil) please also put the pathogens scientific name in parenthesis after the disease. This way, plant pathologists working with related, but not identical, organisms will know what organism the author refers to.

Do not use "I" and "we", keep the writing the past tense and third person when possible. i.e. instead of "In our recent papers, we..." use "Prior research has shown..." or "the author" instead of "I".

Benthiavalicarb is misspelled in the title.

I believe the authors grouped experiments by organism to try and clarify what they did, however, I found this to be difficult to follow. I think grouping organisms by experiment would shorten the paper and provide additional clarity. Many of the experiments were identical, but performed on different plant pathogens/host combinations. I see no benefit in reiterating the method used, if it was identical between basil, tomato and their associated pathogens, for example.

When referring to sporangia density used, please use scientific number format. i.e. 5000 sporangia should be 5x10^3. Some experiments did not have a sporangial concentration stated, that too should be fixed.

I think the authors using mefenoxam resistant isolates was a great choice. It adds to the importance of their work using alternate chemistries.

Overall, I want to commend the author for the extensive work they did. It was obviously a large and robust project and would make a great addition to expand the literature on Oxathiapiprolin/Benthiavalicarb sensitivity in oomycetes, once the grammatical and clarity/cohesiveness issues are addressed.

i would be happy to review this paper again, once these issues are addressed.

Reviewer #2: Please create for experiments conducted in growth chamber and fields

Expt# Crop diseases Fungicides tested Application rate of fungicides

Corrections in the attached file

6. PLOS authors have the option to publish the peer review history of their article (what does this mean?). If published, this will include your full peer review and any attached files.

Reviewer #1: No

Reviewer #2: No

---

## [Decision Letter · Decision Letter 1]

23 Dec 2019

Root treatment with oxathiapiprolin, benthiavalivarb or their mixture provides prolonged systemic protection against oomycete foliar pathogens

PONE-D-19-24369R1

Dear Dr. Cohen,

We are pleased to inform you that your manuscript has been judged scientifically suitable for publication and will be formally accepted for publication once it complies with all outstanding technical requirements.

With kind regards,

Binod Bihari Sahu, Ph.D.

Academic Editor

PLOS ONE

Additional Editor Comments (optional):

Reviewers' comments:

Reviewer's Responses to Questions

**Comments to the Author**

1. If the authors have adequately addressed your comments raised in a previous round of review and you feel that this manuscript is now acceptable for publication, you may indicate that here to bypass the “Comments to the Author” section, enter your conflict of interest statement in the “Confidential to Editor” section, and submit your "Accept" recommendation.

Reviewer #1: All comments have been addressed

2. Is the manuscript technically sound, and do the data support the conclusions?

Reviewer #1: Yes

3. Has the statistical analysis been performed appropriately and rigorously? 

Reviewer #1: Yes

4. Have the authors made all data underlying the findings in their manuscript fully available?

Reviewer #1: Yes

5. Is the manuscript presented in an intelligible fashion and written in standard English?

Reviewer #1: Yes

6. Review Comments to the Author

Reviewer #1: (No Response)

7. PLOS authors have the option to publish the peer review history of their article (what does this mean?). If published, this will include your full peer review and any attached files.

Reviewer #1: No

---

## [Editor Report · Acceptance letter]

31 Dec 2019

PONE-D-19-24369R1 

Root treatment with oxathiapiprolin, benthiavalicarb or their mixture provides prolonged systemic protection against oomycete foliar pathogens 

Dear Dr. Cohen:

I am pleased to inform you that your manuscript has been deemed suitable for publication in PLOS ONE. Congratulations! Your manuscript is now with our production department. 

With kind regards,

on behalf of

Dr. Binod Bihari Sahu 

Academic Editor

PLOS ONE